# Photoactivated Chromophore Corneal Collagen Cross-Linking for Infectious Keratitis (PACK-CXL)—A Comprehensive Review of Diagnostic and Prognostic Factors Involved in Therapeutic Indications and Contraindications

**DOI:** 10.3390/jpm12111907

**Published:** 2022-11-16

**Authors:** Ileana Ramona Barac, Andrada-Raluca Artamonov, George Baltă, Valentin Dinu, Claudia Mehedințu, Anca Bobircă, Florian Baltă, Diana Andreea Barac

**Affiliations:** 1Department of Ophthalmology/ENT, Faculty of Medicine, ‘Carol Davila’ University of Medicine and Pharmacy, 050747 Bucharest, Romania; 2Bucharest Emergency Eye Hospital, 030167 Bucharest, Romania; 3Department of Obstetrics and Gynecology, Faculty of Medicine, ‘Carol Davila’ University of Medicine and Pharmacy, 050747 Bucharest, Romania; 4Department of Rheumatology and Internal Medicine, Faculty of Medicine, ‘Carol Davila’ University of Medicine and Pharmacy, 050747 Bucharest, Romania; 5Faculty of Medicine, ‘Carol Davila’ University of Medicine and Pharmacy, 050747 Bucharest, Romania

**Keywords:** infectious keratitis, corneal ulcer, Photoactivated Chromophore Corneal Collagen Cross-Linking, PACK-CXL

## Abstract

Infectious keratitis is a severe infection of the eye, which requires urgent care in order to prevent permanent complications. Typical cases are usually diagnosed clinically, whereas severe cases also require additional tools, such as direct microscopy, corneal cultures, molecular techniques, or ophthalmic imaging. The initial treatment is empirical, based on the suspected etiology, and is later adjusted as needed. It ranges from topical administration of active substances to oral drugs, or to complex surgeries in advanced situations. A novel alternative is represented by Photoactivated Chromophore Corneal Collagen Cross-Linking (PACK-CXL), which is widely known as a minimally invasive therapy for corneal degenerations. The purpose of this review is to identify the main diagnostic and prognostic factors which further outline the indications and contraindications of PACK-CXL in infectious keratitis. Given the predominantly positive outcomes in the medical literature, we ponder whether this is a promising treatment modality, which should be further evaluated in a systematic, evidence-based manner in order to develop a clear treatment protocol for successful future results, especially in carefully selected cases.

## 1. Introduction

Infectious keratitis is defined as a pathological process in the cornea, caused by the presence of one or more pathogenic microorganisms. It is a medical emergency and if left untreated, it can lead to many complications, including corneal thinning, scarring, perforation, endophthalmitis, loss of sight, or even loss of the eye [1,2]. Unfavorable prognosis is suggested by an ulcer involving the visual axis, the presence of a large infiltrate, as well as low visual acuity at initial work-up, especially in the elderly [3]. The outcome depends on establishing a prompt, correct diagnosis, and then on choosing the most suitable intervention according to the causative agent [4].

Etiology can be either microbial (with bacteria, fungi, protozoa) or viral, and is pivotal to deciding the right treatment plan [5]. Initial therapy is empiric, but absolutely mandatory, because exacerbation is imminent, and diagnostic confirmation usually takes at least 48 h—wasted time being linked to a worse prognosis [1]. Depending on the causal agent, medical management includes topical administration of appropriate anti-infective drugs, cycloplegics, as well as subconjunctival or intrastromal injections, and oral therapy in selected cases. Surgical options consist of corneal biopsy, conjunctival flap, amniotic membrane transplantation, and therapeutic keratoplasty, and are reserved for advanced situations [1,6].

A viable therapeutic alternative is represented by Photoactivated Chromophore Corneal Collagen Cross-Linking (PACK-CXL), a procedure based on the properties of ultraviolet-A (UV-A) light radiations to photoactivate riboflavin (vitamin B2), leading to both stiffening of the corneal stroma and inactivation of pathogens [2,7].

## 2. Materials and Methods

The purpose of this paper is to comprehensively review the medical literature on the specific diagnostic and prognostic factors which dictate the indications and contraindications of PACK-CXL in infectious keratitis, while tracing the history, development, and outcomes of this procedure over the last 15 years—from its first implementation until 27 July 2022.

Although corneal cross-linking represents an already recognized tool in ophthalmology, it has not been widely established in general clinical practice yet, and it is still under evaluation by the scientific community in the treatment of corneal infections. Therefore, we believe such reviews are necessary until guidelines are thoroughly defined and applied.

In this regard, our search was conducted on PubMed and Google Scholar, on 27th of July 2022, using the following keyword: ‘((corneal OR collagen) cross-linking OR photoactivated chromophore OR CXL OR PACK-CXL) AND (keratitis OR corneal ulcer)’. We used both MeSH (PubMed) and free-text vocabulary. In order to be as comprehensive as possible, we included case reports, case series, clinical and epidemiological studies, reviews, and meta-analyses dating as early as 2008, with significant impact, clear methodology, and direct relevance to the subject, as appreciated by the authors of this paper (Figure 1). We considered all articles available in English concerned with the assessment of PACK-CXL in the context of infectious keratitis. We excluded articles about non-infectious keratitis; keratitis caused by CXL; non-infectious corneal melting or ulcer; general reviews about keratitis with brief mentioning of CXL; general reviews about cross-linking or the use cross-linking procedure in the context of various corneal ectasias and other keratopathies; combined CXL and surgical procedures; and cases pertaining to veterinary medicine. Of the 337 papers screened, 55 have been included (Table A1).

## 3. Results

### 3.1. The Challenge of Accuracy

The precise recognition of the pathogen is a process in itself and comprises multiple steps. Clinical examination and microbiological techniques play a central role, but newer investigation methods are emerging and could further complete the diagnostic picture [1]. However, it is important to note the fact that the standard approach is sometimes flawed, considering that observable features do not always follow classical descriptions, that they might be altered or combined in the presence of multiple microorganisms, that cultures take a certain time to grow, and that microbiological detection is not guaranteed, given the high percent of false negatives. Fortunately, the more severe cases, characterized by larger lesions with higher microorganism densities, lead to greater positivity rates in both smears and cultures [4]. Even so, findings in the medical literature indicate that more than 20% of the cases are polymicrobial and more than 50% do not have a positive culture identification of an infectious agent [6].

Upon presentation, previous ocular history might be suggestive of the cause. Contact lens wear is frequently linked to a bacterial (Pseudomonas aeruginosa), fungal, or parasitic infection (Acanthamoeba), and corneal lacerations with vegetal matter (i.e., branches, leaves) may be complicated by a fungal infection [4]. A recent LASIK intervention points towards certain atypical pathogens, including fungi (e.g., Exophiala dermatitidis), non-tuberculous mycobacteria (e.g., Mycobacterium chelonae), and other bacteria (Nocardia, but also methicillin-resistant Staphylococcus aureus or Streptococcus spp.) [6]. However, symptoms are variable and rather non-specific, including decreased vision, discharge, photophobia, tearing, and foreign body sensation with inconsistent levels of pain.

Slit-lamp examination usually reveals infiltrate(s), haze, edema or ulceration, with particular characteristics depending on the cause. Typically, the infiltrate is localized, round-shaped and well-defined in Gram-positive infections, but it is suppurative and rapidly progressing in Gram-negative cases [1]. If it is determined by filamentous fungi, it may appear as an elevated and dry lesion or as an endothelial plaque, but if it is due to Candida, it may be a slowly evolving stromal lesion. Notably, herpetic ulcers are dendritic or geographic, whereas Acanthamoeba induces irregularities in the epithelium [6]. The stromal infiltrates caused by Gram-positive bacteria and Candida are distinct and white-grey in color, whereas those caused by Gram-negative bacteria and Acanthamoeba usually form an immune ring, and those by filamentous fungi have feathery borders and multiple satellite lesions (the later also suggesting the appearance of Acanthamoeba infiltrates) [1]. Gram-positive keratitis presents with minimal haze, in contrast to Gram-negative cases, whereas filamentous fungus infections develop endothelial plaques. Moreover, there are individual characteristics pertaining to some species. Nocardia determines white, numerous infiltrates, arranged in a wreath, with accompanying fine filaments towards healthy cornea. On the other hand, non-tuberculous mycobacteria keratitis has been described as having a ‘cracked windshield’ appearance. Microsporidium can be mistaken for atypical adenoviral keratoconjunctivitis, due to its punctate epithelial lesions and subepithelial scarring [6]. Viral keratitides are the most heterogeneous. The herpes simplex virus itself determines either epithelial, stromal, endothelial or kerato-uveitic presentations, whereas the varicella-zoster virus typically manifests as nummular keratitis [1].

Despite their detailed descriptions, these findings are not consistent and can be unreliable in clinical settings, as mentioned above. It has been shown that cornea-trained specialists correctly distinguish bacterial keratitis from fungal keratitis in 66% of the cases [8]. This is why the objective identification of the etiology through microbiology techniques is oftentimes employed, and primarily consists of staining and microscopy examination, cultures, and sensitivity testing [4]. Theoretically, this should represent the protocol in all suspected infections [5], but most often, clinicians, through no fault of their own, resort to treating the majority of cases empirically, based on experience. This is in part due to the fact that the actual usefulness of additional diagnostic steps remains a controversial topic, taking into consideration that they are more expensive and time-consuming, and that broad-spectrum antibiotics often lead to good outcomes, especially in small, superficial, peripheral ulcers of non-traumatic origin [9]. Certain guidelines state that such investigations are only needed in advanced cases of keratitis, which involve the visual axis, consist of large or multiple lesions, display atypical characteristics, appear in the context of recent corneal surgery, or do not respond to broad-spectrum therapy [1], and several papers have underlined that they are actually employed in only 5% to 15% of the cases [4].

In order to identify the pathogen, the first part is represented by corneal scrapings under topical anesthesia, ideally after removing mucus and debris, primarily targeting the lesion’s base and active borders [4]. Then, samples are smeared onto slides stained with various substances and analyzed using direct microscopy. The most widely used stains are Gram—for bacteria, Giemsa or potassium hydroxide (KOH)—for fungi, and Calcofluor White Staining (CFW) or lactophenol for Acanthamoeba [9]. Gram is common and well-standardized in classifying cocci and bacilli, but errors due to both logistics and interpretation generate vast variability between laboratories, with reported sensitivity ranging from 30% to 100%. In addition, there can remain unstained bacteria, such as Mycobacterium spp. or Nocardia spp. Therefore, Ziehl-Nielsen (ZN) or Kinyoun (modified ZN) could be necessary in selected cases [4]. On the other hand, yeasts, including Candida, might be missed under Giemsa, or mistaken for artifacts [9].

The second use of corneal scrapings material is for cultures, which is the definitive gold standard in current medical practice and should always include appropriate media for both bacteria and fungi [4,9]. Apart from corneal scrapings, materials such as contact lenses, their cases and cleaning solutions, or loose sutures on the eye surface, which have been in direct contact with infected tissues, can also be cultured [4]. The positivity rate varies greatly, similar to direct microscopy, and is influenced by multiple factors, such as technical difficulties, delays, low pathogen load, clinical severity, or recent use of anti-infective agents or of topical steroids [4,5,6]. In addition, pathogens have to be differentiated from commensal microorganisms, using a multitude of criteria. Logistics include a variety of well-defined protocols which can be employed, and they include direct and indirect inoculations on solid agars or liquids [4]. For common bacteria, usual choices encompass blood agar, chocolate agar, or brain–heart infusion at 37°, whereas Mycobacterium grows on Lowenstein–Jensen and Middlebrook media. Fungi can develop in similar conditions if antibacterials are added, or can grow on Sabouraud dextrose agar. Acanthamoeba needs non-nutrient agar, with an E. coli overlay [1,6]. Then, these media are incubated in strict conditions for 1–21 days, and are re-examined daily for noticeable changes [9]. Typical bacteria grow relatively fast, but Nocardia, Mycobacterium, fungi, and Acanthamoeba require significantly more time, which can sometimes be unpredictably long [4].

Another point of discussion is concerned with antibiotic sensitivity and resistance, as reactionary patterns to the readily available ophthalmic drugs are changing permanently [9]. However, laboratory methods determine systemic concentrations, rather than topical ones, which can be confusing, as they do not take into consideration the direct route of administration, the frequency of instillations, or the fortified forms used in ophthalmology. Therefore, it is advisable that the treatment course should be planned by integrating both microbiological data and timely assessments of the patients’ clinical evolution [4].

All the aforementioned information is directed towards the main diagnostic proceedings in microbial infections, i.e., bacterial, fungal or parasite cases. As far as viral keratitis is involved, diagnosis is usually clinical, given the particular appearance at the slit-lamp and the additional presence of herpetic vesicles [1]. The arsenal for objective confirmation is fundamentally different from that of other types of pathogens, and comprises of cultures on Vero cell lines, direct antibody identification using fluorescence, and Polymerase Chain Reaction (PCR) from active lesions or tears samples [5,6].

Consequently, the most common pathogens involved in keratitides are the following:Gram-positive bacteria: coagulase-negative staphylococci, Staphylococcus aureus, Streptococcus pneumoniae;Gram-negative bacteria: Pseudomonas aeruginosa, Enterobacteriaceae, Moraxella, Haemophilus, Neisseria gonorrhoeae;acid-fast bacteria: Mycobacterium, Nocardia;filamentous fungi: Fusarium, Aspergillus, Curvularia, Alternaria;yeasts: Candida albicans, other Candida spp., Cryptococcus;fungi-like: Microsporidium;parasites: Acanthamoeba;viruses: HSV 1, CMV. VZV, Adenovirus [1,4,6].

In progressing non-responsive cases with previous sterile cultures, additional diagnostic techniques are employed. Corneal biopsy might be performed in order to gain deeper access to corneal infiltrates and to facilitate histopathology analyses, apart from direct microscopy and culturing, which clarifies a potential fungal or Acanthamoeba etiology. However, clinicians should be wary of perforation risks, especially with thinning, melting, or necrotic tissue [4]. Similarly, impression cytology could be used for the same purposes, as it is especially helpful for fungi or parasite cases [6], whereas Transmission Electron Microscopy (TEM) represents the gold standard for confirming the presence of Microsporidium spores [1].

Moreover, PCR can also be used in all microbial infections, with faster results and high sensitivity, including pathogens with slow and difficult growth on cultures. Its disadvantages include high equipment and training costs, the need to target microorganisms specifically by picking the right primers, and lower specificity, as it does not discriminate active from dead pathogens, or from background flora [1]. An alternative molecular diagnosis method is represented by mass spectrometry, which might play an important role in identifying rare species [4]. Over time, more sophisticated techniques have been developed, such as metagenomics next-generation sequencing (NGS), which is based on nucleic acid amplification and shares similar working principles with PCR but is currently more suitable to research than to clinical use [1].

Novel ophthalmic non-invasive imaging tools have been proposed within diagnostic frameworks. For this scope, in vivo confocal microscopy (IVCM) has been used to generate real-time images with a resolution of 1 μm, which is sensitive enough in order to identify fungal filaments or parasitic cysts—but not smaller pathogens—anywhere in the depth of the cornea. However, it is expensive, heavily operator-dependent, and it generates potential errors due to artifacts [1,9]. Anterior Segment Optical Coherence Tomography (AS-OCT) is a more accessible method, with excellent anatomical and pathological descriptions, as well as precise quantitative data. It can be used in conjunction with biomicroscopy, when deep ulcers or large infiltrates cannot be assessed properly and obscure deeper ocular tissues. It offers valuable information on corneal thickness, which has been shown to fluctuate proportionately with inflammation severity, and can therefore be useful in monitoring progress, but cannot identify the responsible pathogen [4,10].

Lastly, Artificial Intelligence shows immense promise in analyzing images, with accurate interpretations through the means of well-trained pattern recognition algorithms. Thus, it can differentiate active lesions from corneal scars, or typical bacterial and fungal ulcers. For the moment, the most important real-life utility could be the rapid screening, diagnosis, and appropriate recommendation-making based on external photos of the eye, in the frame of a telemedical service for communities with low access to an ophthalmologist [1].

### 3.2. The Challenge of Choice

It is important to underline the fact that successful treatment of infectious keratitis is linked to the accurate identification of the responsible pathogen [4].

Therefore, typical treatment in bacterial cases consists of broad-spectrum antibiotics with topical administration, which should be initiated empirically as soon as possible—eventually, after collecting appropriate samples for laboratory analyses, if indicated. Gold standard schemes include either fluoroquinolone monotherapy, or fortified combinations of cephalosporins and aminoglycosides [1]. The subconjunctival route might be preferred in specific cases, when the risk of perforation is high or adherence issues might occur, whereas systemic therapy is necessary if the infection keeps spreading towards adjacent tissues. The two main medical alternatives have been shown to be similar in efficacy, and the choice depends on clinical judgment, experience of the physician, and drug availability [11]. However, growing resistance towards antibacterial substances represents a major public health issue, posing a challenge in specific cases, especially those of staphylococcal or Pseudomonas origin. The impact of this phenomenon is currently not as defined in ophthalmology as it is in systemic infections, but prudence is advisable in order to prevent a future decline in susceptibility to common antibiotics, especially in typical cases that are relatively straightforward to treat in today’s climate [5].

On the other hand, fungal keratitis is medically cured with antimycotic therapy, but generally has a worse prognosis compared to bacterial ulcers, due to lower drug penetration levels, as well as diagnosis difficulties [1]. Topical administration includes several alternatives, such as 5% natamycin or 1% itraconazole for filamentous fungi, and 0.15% amphotericin B or fluconazole for yeasts [6]. Intracameral or intrastromal injections are useful in extended or non-responsive infections [1], and various oral triazoles are also available [6].

An even more complicated situation is represented by Acanthamoeba infections, as timely diagnosis and aggressive treatments are fundamental to satisfactory clinical evolution. Combinations of medicine must be used, because an agent capable of eliminating both trophozoites and cysts does not exist [1]. However, dibromopropamidine, hexamidine, chlorhexidine, and polyhexamethyl-biguanide are acceptable topical options [6].

As far as viral keratitis is concerned, treatment remains controversial to some degree and depends on the site of the infection. If only the corneal epithelium is affected, topical therapy with acyclovir should suffice. However, if there is stromal involvement, topical steroids are mandatory, with careful monitoring of local complications, especially a rise in IOP [1,6].

Regardless of the cause, adjunctive therapy plays an additional role in catalyzing the good clinical outcome of all cases. Regular debridement and saline instillations help removing necrotic tissue and secretions, thus decreasing local pathogen load and increasing treatment penetration [12]. Cycloplegics are prescribed in order to decrease synechiae development and pain when there is remarkable inflammation in the anterior chamber. On the other hand, the role of topical steroids in bacterial cases is unclear, the risk-benefit balance is not yet calibrated by definitive evidence, and its use should be judicious and dictated by case particularities, such as visual axis involvement, and an already good response to antibiotic therapy [11].

Even with prompt medical treatment, results often remain poor due to complications, and severe cases can lead to rapid deterioration, with no alternative other than surgical treatment. This includes amniotic membrane transplantation and penetrating keratoplasty [1,6]. However, operating on an infected eye is especially risky and prone to complications and failure, so that evisceration and enucleation are indicated in extreme cases where the visual potential has been lost [12].

### 3.3. The Challenge of Novelty

In recent years, an alternative therapy has been proposed for infectious keratitis, represented by Corneal Collagen Cross-Linking (CXL). It has already been used with good outcomes, as an adjuvant to antimicrobial therapy in patients with treatment-resistant corneal infections, bacterial-only [13,14,15,16,17], fungal-only [18,19,20], in mixed bacterial-fungal cases [2,21,22,23,24,25], or in Acanthamoeba infections [26,27,28]. Some investigators focused exclusively on refractory corneal ulcers and found that cross-linking therapy is beneficial [29,30,31,32], and that it can also markedly reduce healing time [33].

The effectiveness of CXL as a primary therapy has been shown both in animal models [34,35], and in clinical studies [36], but other authors argued that this approach has no advantage over the standard treatment [37], and a recent randomized clinical trial underlined that the clear benefit of CXL per primam therapy could not be yet proven, despite a lower complication rate in the cross-linking-only group [38].

The advantages of this minimally invasive approach are reinforced by a study focused on the one-year follow-up after CXL, which confirms the favorable prognosis long-term post-procedure [39].

CXL uses riboflavin (vitamin B2) drops, which act as a chromophore, photoactivated by UV-A radiations at 365–370 nm wavelength. Thus, free radicals are generated, promoting both an increase in collagen fiber diameter, as well as the creation of additional collagen-proteoglycan bonds in the corneal stroma. These new cross-links increase biomechanical stiffness, useful in the treatment of ectatic disorders of the cornea, markedly keratoconus, which represents the most widely-known scope of this minimally-invasive medical procedure [7,40,41].

In 2003, Wollensak, Sporl, and Seiler reported the preliminary successful results of CXL in stopping keratoconus progression, in a non-randomized five-year study, using a specific set of steps which would later be called “the Dresden protocol”: under sterile conditions in the operating room, the central corneal epithelium is removed under local anesthesia; 0.1% riboflavin solution is applied every 5 min for 30 min; then, 370 nm UV-A irradiation begins, using a lamp, 1 cm away from the cornea, for 30 min, with an intensity of 3 mW/cm^2^, translating to a total amount of energy of 5.4 J/cm^2^; finally, a bandage contact lens is applied. Topical antibiotic drops are administered until reepithelialization is noted [42]. In accordance with the Bunsen–Roscoe Law of Reciprocity of Photochemistry, which asserts that the effects of UV-A radiation and the final dose of energy are directly correlated, regardless of the combinations between illumination time and intensity of light, as long as the total amount of energy remains the same [43], modifications to this standardized model have been proposed. In order to improve efficiency in this rather time-consuming technique, ophthalmologists employed UV-A rays of higher intensity, thereby shortening the procedure and consequently decreasing the chance of corneal dehydration, while increasing the comfort of both patient and doctor, with beneficial results without supplementary risks [44].

Furthermore, an additional role of riboflavin in combination with UV-A light includes pathogen inactivation, by inducing DNA damage in bacteria and viruses, to a degree which appears to make it more difficult to repair (compared to degradation suffered by host cells) [45], and also by increasing resistance against protein digesting enzymes, such as collagenase, trypsin, and pepsin (similar to the metalloproteinases involved in corneal ulcers) [46]. Clinically, this application has been demonstrated in transfusion medicine, by microbial decontamination of blood products [47,48]. Interestingly, an initial variant of the cross-linking procedure had actually been successfully employed by Schnitzler, Spoerl, and Seiler in 2000, in the treatment of non-infectious corneal melting, even before the publication of the classical Dresden protocol for CXL in keratoconus [42,49]. Furthermore, in 2008, Iseli et al. conducted the first study to assess the efficacy of CXL in treatment-resistant microbial keratitis, with promising results [50], and in 2012, Makdoumi et al. performed a pilot study in order to evaluate CXL as primary therapy in bacterial keratitis, with no prior antibiotic administration, with yet another favorable assessment [36].

Accelerated protocols have also been employed (either 9 mW/cm^2^ for 10 min, 18 mW/cm^2^ for 5 min, 36 mW/cm^2^ for 2.5 min), all reaching the same positive conclusions towards a more efficient technique [51,52,53,54]. Further studies might bring other optimized techniques, such as the positive link between higher UV fluence and increased levels of microbial killing [28,55], as well as higher concentration of riboflavin [56].

Following all these advancements, at the 2013 meeting of the International Congress of Corneal Cross-Linking in Dublin, a clear difference has been established between the treatment technique in keratoconus (from then on, simply called Corneal Cross-Linking or CXL) and the treatment of infectious keratitis (from then on, called Photoactivated Chromophore for Infectious Keratitis or PACK-CXL) [7].

Additional indisputable advantages are represented by the fact that PACK-CXL does not further contribute to antibiotic resistance, as corneas are considered sterile after the procedure. Along with the financial benefits, there were reductions across three areas: cost of medication itself, number of follow-ups, and potential hospitalizations [25,52]. Another interesting finding is that, apart from the great clinical feedback, there is electrophysiological proof that cross-linking does not damage the retina and the optic nerve [57].

However, success rates might be unequal among pathogen types, with Gram-negative bacteria being the most susceptible and fungi, the least [58]. Antibiotic resistance does not seem to be correlated with photooxidative stress resistance [59]. One recent meta-analysis underlined rigorous evidence in case of bacterial infection, suggesting further inquiries are needed for those of fungal, parasite, or viral origin [60]. On the other hand, other reviews raised concerns about the deep clinical and methodological heterogeneity of the available literature, and actually refrained from drawing definitive conclusions [61,62].

Among potential complications of PACK-CXL, a meta-analysis noted corneal edema, loss of endothelial cells, and disease progression with decompensation, leading to perforation in rare cases. This might be influenced by the corneal depth of the infiltrates, as more than 250 microns increase both the risk of endothelial cell loss and resistance to the effect of PACK-CXL [63,64]. It is essential to take note of the fact that the procedure itself could facilitate infections, considering that the step of epithelial debridement removes the physical barrier of the cornea, thus exposing it to contagious agents [65]. Moreover, this procedure is cytotoxic to keratocytes, up to 300 microns, but especially in the first 100 microns, which absorb half of the total energy [66]. A major observation is the reactivation of previous herpes simplex keratitis following CXL, which makes it a contraindication [24,63], thus illustrating the importance of an accurate diagnosis.

Additional concerns have been expressed regarding the efficiency of PACK-CXL as primary treatment in fungal keratitis, as it might not lead to expected outcomes in deep stromal infections [58,67,68], by either not showing clinical advantages over conservative therapy [69,70], or even by leading to worse results when compared to medication [71,72]. However, these negative outcomes have been highly debated by scientists in the field [73,74]. Therefore, conclusions remain unclear.

Important differences have also been found among re-epithelization periods for different microorganisms. Whereas ulcers of bacterial origin can heal in as fast as 3 days, fungal and protozoa keratitides can take much longer (up to more than 100 days). Among bacteria, Mycobacterium is the most problematic, also needing more than 100 days for the corneal wound to close [63].

On the other hand, a recent randomized, prospective, phase 3 trial published in 2022 by Hafezi et al. underlined no difference in major complications between medically treated patients and the cross-linking group [25]. Given the complexities of these conflicting literature findings, more inquiries are needed in order to assess these problematic aspects, as already suggested by previous studies [66,75].

## 4. Discussion

It is known that infectious keratitis is a potentially dangerous condition, in which the precise diagnosis and accurate treatment are paramount. In addition, appropriate timing is of utmost importance in order to prevent eye-threatening complications. However, the gold standard therapy is well-established, and it renders good results in a majority of cases. Therefore, we raised the question if there is a place for PACK-CXL in everyday medical practice by identifying its possible indications and contraindications, from diagnostic and prognostic points of view.

As shown above, the medical literature confirmed the efficacy of PACK-CXL in a variety of case series and small studies conducted by multiple physicians throughout the world. Consequently, we attempted to identify a string of common particularities in a majority of these cases, regarding diagnosis difficulties (Table 1).

In addition, we also underline a variety of prognostic factors, shared among many of the situations when cross-linking was used (Table 2).

Moreover, by applying similar reasoning, we propose several scenarios when PACK-CXL could be indicated (Table 3).

However, contraindications of this procedure are not as clear, and neither is their absolute or relative nature. Some of them can be derived from the exclusion criteria of the initial study of safety of CXL in corneal dystrophies [76] (Table 4).

We must acknowledge the fact that this randomized clinical trial, which led to the FDA approval of Corneal Collagen Cross-Linking in the treatment of keratoconus, only excluded certain patient categories, considered at risk for complications or therapeutic failure. It has not explicitly demonstrated that these criteria also translate to contraindications. For instance, later studies extended the age range to 8-year-olds [43], and further input is needed in order to confirm each criterion and to define even more precise parameters, based on indisputable evidence.

Other contraindications of PACK-CXL in the context of corneal ulcers, as mentioned above in the review section, are keratitis of a viral cause that infiltrates deeper than 250 microns.

Considering the current medical climate, which has been built on evidence over the last decades and is dominated by protocols, a multitude of reservations about PACK-CXL occur, halting its large adoption in the treatment of microbial keratitis, at least for the moment. Some reasons for this reluctance include the lack of systematic research and the lack of official approval from international regulatory organisms. These two main arguments are doubled by occasional publication of papers with questionable research methodologies, heterogeneous results, lack of reproducibility, and surrounding controversy, which makes conclusions harder to draw. This uncertainty is especially relevant when the cross-linking procedure comes into discussion for the most difficult and advanced cases, where the treatment course has to be chosen carefully in order to prevent the imminent ocular damage.

It is still unclear if PACK-CXL should be used per primam, without recent history of medical treatment failure, or in relatively mild cases. Moreover, it has not yet been tested in certain populations, such as young children or pregnant patients, which might benefit from it (at least in comparison to surgery). However, it is important not to fall into the trap of picking a novel procedure for the sake of novelty only, especially until safety is assured. There are still many unanswered questions: Should this be an adjuvant to antimicrobial treatment, or be used per primam? What would the indications and contraindications be in relation to corneal thickness? What kind of protocol (standard, intermediate, fast) is best suited? What kind of riboflavin should be used to maximize results? How many times can the procedure be repeated? Under which conditions should patient from vulnerable populations receive this therapy?

The strengths of this paper consist of a thorough inquiry of available research information, culminating in the most cohesive list of potential indications and contraindications of PACK-CXL in infectious keratitis at the moment (as far as we managed to find through our review), based on critical thinking and appraisal; a well-defined keyword and search framework, facilitating a precise exploration of medical literature on the chosen subject, suitable for the academic purpose of this investigation; the large number of screened articles, which helped to trace the development of cross-linking back to its inception; and a comprehensive review of the diagnostic and prognostic factors of infectious keratitis, and an extensive description of CXL, including history, working mechanisms, various protocols, and results and weaknesses, which creates a greater perspective on the topic.

The limitations include the narrative nature of the review, which does not comply with PRISMA guidelines and is prone to subjectivity issues; the lack of in-depth mentions, analyses, and comparisons of methodology for the papers included in this review; and the absence of statistical confirmation for our descriptive observations and qualitative conclusions.

## 5. Conclusions

PACK-CXL represents a promising treatment for microbial keratitis, with multiple significant advantages—it is not influenced by the type of pathogen and its characteristics (including resistance to medication), it is minimally invasive, easy and safe to perform, and it leads to great clinical and imaging outcomes, with decreased costs and increased patient comfort. Over the following years, it could become an effective and well-established tool in battling difficult cases, minimizing or even removing the need for surgery, as well as preventing other serious sequelae, such as corneal perforation. It is believed to represent an effective, safe alternative to traditional medical treatment, yet more systematic research is needed in order to establish the exact indications and the specific protocols.

## Figures and Tables

**Figure 1 jpm-12-01907-f001:**
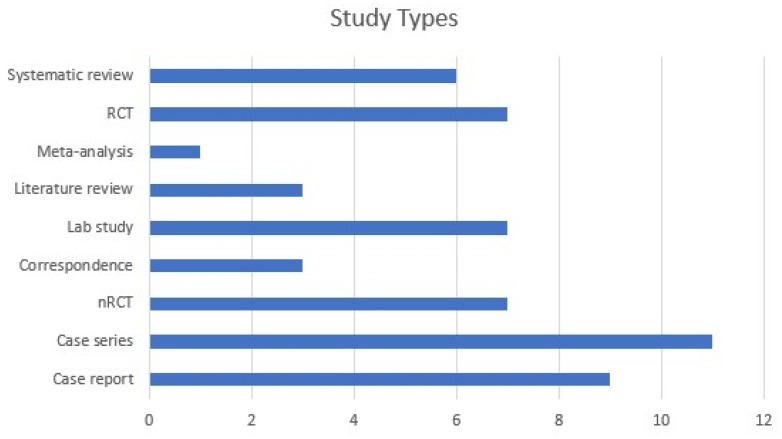
Bar graph showing distribution of study types selected for review. (RCT: randomized controlled trial, nRCT: non-randomized clinical trial).

**Table 1 jpm-12-01907-t001:** Relevant causes for diagnostic failure.

Diagnostic Difficulties
Repeated sterile cultures, either caused by unsuitable previous use of anti-infective medicine, or by the supposed presence of fastidious or rare pathogens
Polymicrobial infections, in which some of the responsible agents are not identified
Lack of access to advanced diagnostic techniques, either caused by lack of funding, available technology, or trained specialists

**Table 2 jpm-12-01907-t002:** Clinical scenarios where CXL was employed.

Clinical Scenarios
Severe, advanced cases, with late presentation
Cases non-responsive to usual therapy, progressive despite correct medical treatment
Infections involving the visual axis
Cases with ominous signs of imminent complications

**Table 3 jpm-12-01907-t003:** Possible PACK-CXL indications.

PACK-CXL Indications	Reasoning
Polymicrobial infections, even if not all of them have been identified	To reduce treatment costs, to improve adherence and, ultimately, to spare the patient from the exposure to multiple potent drugs and their possible adverse effects
Documented resistance to the available anti-infective agents, or remarkable shifts in local susceptibility patterns	To obviate potential future issues in the community
Corneal ulcers following trauma with significant contamination	To reduce microbial load as quickly as possible
Patients with severe keratitis and monocular vision	To reduce microbial load as quickly as possible
Allergies, sensitivity, or contraindications to the recommended medical therapy	To help preserve the ocular surface and to reduce the inflammatory response
History or suspicion of poor adherence	To reduce the need for long-term therapy
Vulnerable populations (pregnant women, elderly patients)	for whom potent systemic therapies or surgeries could be detrimental

**Table 4 jpm-12-01907-t004:** CXL contraindications.

PACK-CXL Contraindications
Allergies, sensitivity, or contraindications to riboflavin, to local anesthetics, or to any other materials used during the procedure
Corneal thickness of less than 375 microns, before debridement of the epithelium
History or likelihood of delayed corneal wound healing
Significant corneal scarring or opacification
History of viral keratitis
Aphakia, or pseudophakia with a non-UV-blocking lens
Nystagmus, or any disorder which might interfere with a steady gaze
Pregnancy or nursing
Age under 12 years old

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
