# Peer review of "Photoactivated Chromophore Corneal Collagen Cross-Linking for Infectious Keratitis (PACK-CXL)—A Comprehensive Review of Diagnostic and Prognostic Factors Involved in Therapeutic Indications and Contraindications"

_jpm, 2022, doi:10.3390/jpm12111907_

Round 1
Reviewer 1 Report (New Reviewer)
Very interesting manuscript, well written with great detail. It provides an overview of corneal pathologies and the use of CXL for treatment in a simple and clear way. This is a good comprehensive review article.
Author Response
Thank you so much for your kind comments.
Reviewer 2 Report (New Reviewer)
This is a really interesting review concerning the use of Photoactivated Chromophore Corneal Collagen Cross-linking in the treatment of infectious keratitis.
This review is well organized and focuses on a very challenging topic, that absolutely needs further investigations.
I suggest the authors to include Tables and/or Figures for a better visualization of the key points of their review. In the present form, the review appears as a "wall of text" and it could be difficult for the readers to read it and to pay attentions on the main aspects of this article.
Furthermore, a minor revision of the English should be performed.
Author Response
Thank you for your comments.
We have added one figure, 3 tables and one Appendix.
The paper has been proofread and adjustments to the language have been made.
This manuscript is a resubmission of an earlier submission. The following is a list of the peer review reports and author responses from that submission.
Round 1
Reviewer 1 Report
Thank you. This is a narrative review looking at the use of PACK-CXL in the treatment of infectious keratitis. This is an interesting topic particularly in the AMR era. In its current format it reads more as a narrative on the clinical presentation, diagnosis and treatment of IK as a whole. I would suggest some major restructuring of the review before consideration for publication. The first 7 pages of the review (up to and including section 3.2) should be condensed and placed in the introduction section. The results section would benefit from a major restructuring and the authors should include more comprehensive details of the studies included so that the reader can more easily relate these to their clinical practice. The results section would benefit significantly from a table stating the population, intervention, study outcomes and follow up times of the studies included in the review. In the discussion section, the authors state they have tried to identify common factors between all the current PACK-CXL studies and propose several interesting scenarios that PACK-CXL could be indicated but this is not related to any evidence that is presented in the results section. It is difficult to illicit what this review in its current form adds in addition to a recently published meta-analysis on the subject (reference 61 in the review).
Reviewer 2 Report
This manuscript is intended to be a review on the diagnostic and prognostic factors that may help aid a clinician’s decision to pursue PACK-CXL for infectious keratitis and also a review of therapeutic indications and contraindications. The first half of the manuscript (up to line 280) is an excellent review of infectious keratitis with minimal mention of PACK-CXL. The next section (lines 281-346) is a good review of CXL. The last section addresses the topic of PACK-CXL in infectious keratitis, divided into two sections—one reviewing the existing literature and another (the Discussion section) providing the authors’ interpretations ideal scenarios for PACK-CXL and contraindications.
The information provided in his manuscript is good, however, the impact of the manuscript could be improved through significant reorganization of the text. The focus of this manuscript is supposed to be on PACK-CXL, especially the topics of the Discussion section, not a general review of infectious keratitis (which occupies the reader for the first half of the manuscript). Perhaps the main points of the discussion can be used as a framework for the entire manuscript so that the focus remains on PACK-CXL throughout.
Overall, the English language of this manuscript is very readable. The authors should carefully scrutinize each sentence to make sure that they are not overstating or overgeneralizing. A few examples are provided here but the entire manuscript should be carefully checked, for example:
Line 47: does amniotic membrane transplantation reduce microbial load? Reference?
Line 100: inconsistent levels of pain: one condition stands out here. Acanthamoeba keratitis can cause excrutiating pain line no other
Lines 308-309: after the drops initially extend 308 throughout the thickness of the cornea for 5 minutes—this statement does not make sense
Line 324: increasing resistance “of the cornea”…
Lines 377-381: does this paragraph refer to with or without PACK-CXL?
Reviewer 3 Report
PACK-CXL considered as alternative and not adjunct to the medical Theraphy.
CXL must not be considered as a minimally invasive therapy
CXL isn’t considered as a potential treatment for Viral Keratitis
The paper doesn’t clearly highlight any inclusion criteria for this procedure furthermore it doesn’t point out any type of emerging strategy to manage a keratitis even from the literature only.
The paper doesn’t highlight as well the different eradication rates of the infection considering the agents involved.